# Yellow Pea Pasta Enhances the Saltiness and Suppression of Postprandial Blood Glucose Elevation

**DOI:** 10.3390/nu15020283

**Published:** 2023-01-05

**Authors:** Yoshihiro Tsuchiya, Joto Yoshimoto, Hiroto Kobayashi, Sho Ishii, Mikiya Kishi

**Affiliations:** Central Research Institute, Mizkan Holdings Co., Ltd., Handa-Shi 475-8585, Japan

**Keywords:** salt, yellow pea, glycemic index, insulin, satiety

## Abstract

Salt and carbohydrates, two causes of elevated blood glucose, are essential components for survival; however, excessive intake of either is a known health risk. In a previous study, we reported the usefulness of pasta prepared from yellow pea (YPP) as a functional staple food that is beneficial for blood sugar control. In this study, we investigated the usefulness of YPP in reducing health risks by examining its effects on saltiness, postprandial satisfaction, and second meal. The results showed that YPP tasted saltier than conventional pasta made from semolina wheat when prepared with a 0.75% salt concentration. In addition, we examined blood glucose levels, insulin secretion, and postprandial hunger over a longer period than in previous studies. We observed that when the same amount of YPP and wheat pasta were eaten, the elevation in blood glucose and insulin secretion was lower after YPP consumption while maintaining a similar level of satiety. Furthermore, YPP was also observed to be able to suppress elevated insulin levels at the second meal.

## 1. Introduction

Although sodium chloride (salt) is essential for humans, excessive intake is known to increase the risk of hypertension, cardiovascular disease, and kidney disease [1,2]. The World Health Organization (WHO) has set a global target salt intake of 5 g per day [3]. According to the Dietary Guidelines for Americans 2010, persons in the United States should limit daily salt sodium intake to <5.8 g [4]. However, in Japan, where salt intake is known to be higher than in other countries due to the food items consumed (10.9 g for men and 9.2 g for women [5]), the Ministry of Health, Labor, and Welfare has set a target salt intake of <7.5 g and <6.5 g per day for men and women over the age of 18, respectively [6]. This background, combined with growing health consciousness in recent years, has led to the development of many foods that aim to reduce salt intake. It is known that the addition of umami, kokumi, and aroma makes the same amount of salt taste saltier and more satisfying [7,8,9]. By applying this phenomenon, we may be able to reduce salt intake more easily without introducing a health burden.

For food to taste good, it needs to have the right amount of saltiness. As staple foods are consumed in large quantities, even small differences in salt concentrations can have a large impact on the total quantity of salt consumed. Therefore, when considering salt reduction as a way to improve eating habits, it is important to eat staple foods with low salt content as much as possible.

On the other hand, staple foods are generally prepared from grains and processed products. Grains contain many carbohydrates that increase glucose concentration in the blood (i.e., blood sugar levels). Blood glucose is an important source of energy for sustaining vital functions; however, excessive elevation or chronic hyperglycemia is known to cause diabetes, coronary artery disease, obesity, and other conditions that threaten people’s health [10]. Therefore, staple foods that do not cause a rapid increase in postprandial blood sugar levels are effective in preventing these diseases. The glycemic index (GI) is used as an indicator of the rise in postprandial blood glucose after eating certain foods. GI is defined as the ratio of the area under the blood glucose curve (IAUC) measured up to two hours after the consumption of a test meal containing a certain amount of carbohydrate to the IAUC of the reference meal, which is 100. Based on glucose, a GI of 70 or higher is defined as a high-GI food, 69–56 as a medium-GI food, and 55 or lower as a low-GI food. The Food and Agriculture Organization recommends a low GI diet as a measure to prevent diseases such as diabetes, coronary artery disease, and obesity. In Japan, it has been reported that a low GI diet is effective for good blood glucose control and the improvement of blood lipids. Wheat pasta (WP) and corn tortillas are low-GI staple foods. In contrast, white rice (WR), white bread, etc., which many people like to eat, are high-GI foods and tend to cause postprandial blood glucose elevation [11]. In addition, some foods, including legumes, are known to have a “second meal effect”, i.e., they influence blood glucose fluctuations in the next meal [12]. These studies suggest that resistant starch, dietary fiber, and other substances ferment in the gut and regulate insulin levels, making it less likely that carbohydrates consumed in the next meal will cause a spike in blood sugar. Eating is not like a single-dose shot; it must be continued intermittently for survival until death. Therefore, tracking blood glucose fluctuations for a longer period beyond the first two hours after a meal and investigating the effects of the meal on blood glucose even after the next meal will provide useful insights for a broader understanding of the effect of food on blood glucose.

Postprandial blood glucose levels and satiety have been examined in several studies, but there is no consensus on this topic. In other words, there is a mixture of studies that report a correlation between blood glucose and satiety and those that do not [13,14]. Foods that make us feel full easily (i.e., foods that make us feel less hungry) are thought to effectively prevent overeating [15].

Our previous research on yellow pea pasta (YPP) showed that it could serve as a low-GI staple food [16]. Based on the background described above, this study was conducted to test whether YPP could be consumed as a functional alternative staple food, bearing in mind the following objectives:(1)We compared the taste characteristics of YPP to commercial salt at various times.(2)We compared the effects of YPP and WP on postprandial blood glucose and insulin levels after a first and subsequent meal.(3)We compared the maintenance of satiety after eating the same of YPP to eating the same weight of WP.(4)By doing these tests, we tested whether YPP is a functional staple that could help reduce salt intake and control blood sugar.

## 2. Materials and Methods

### 2.1. Materials to Be Tested

YPP (ZENB JAPAN Co., Ltd., Aichi, Japan) and WP (Nisshin Foods Inc., Tokyo, Japan), commercially available in Japan, were used in this study. Food-grade salt (Salt Industry Center of Japan, Tokyo, Japan) and olive oil (Nisshin OilliO Group, Ltd., Tokyo, Japan) were purchased and used. Glucose (PEARL ACE CORPORATION, Tokyo, Japan) was dissolved in water and used as a reference for postprandial blood glucose testing. To verify the effects of the second meal, retort rice (Sato Foods Co., Ltd., Niigata, Japan) and seasoning powder containing almost no carbohydrates (Red Shiso Leaf Rice Seasoning^®^; Mishima Foods Co., Ltd., Hiroshima, Japan) were used.

### 2.2. Sensory Evaluation

A research and development staff member, who worked for a product manufacturer and was familiar with the sensory testing of the five basic flavors, served as a panelist to conduct a nine-step sensory evaluation [17], following the same method as Rathod et al. To describe the test method briefly, pasta samples were boiled in salt-free water for 5 min (ratio of 80 g of dry pasta to 1 L). The pasta was removed from the boiling water. Next, 30% salt water was added to 100 g (wet weight) of the boiled pasta to reach the final salt concentration, and 10 g of olive oil was added. The mixture was immediately stirred 30 times for homogenization, and approximately 50 g was instantly provided to the subjects. The seasoned test meal was ready within 5 min of boiling; hence, it was immediately available for evaluation.

Similarly, a sensory test was conducted on the boiled water separated from the pasta on a different day. The boiled water was weighed, salt was added to reach the final concentration, and it was instantly used for evaluation. Respondents rated their perceived saltiness of the YPP on the evaluation form using the following scale: 1 point for “extremely mild”; 2 points for “very mild”; 3 points for “mild”; 4 points for “somewhat mild”; 5 points for “neither intense nor mild”; 6 points for “somewhat intense”; 7 points for “intense”; 8 points for “very intense”; and 9 points for “extremely intense”. In this study, saltiness, umami, sweetness, bitterness, sourness, kokumi, and overall acceptance were evaluated.

### 2.3. Subjects for the Study of Changes in Postprandial Blood Glucose, Insulin Measurement, and Postprandial Satiety

The subjects were healthy men and women between the ages of 20 and 49 years old; with a body mass index (BMI) between 18.5 kg/m^2^ to less than 25.0 kg/m^2^; with no abnormalities in glucose tolerance according to tests conducted in the past year; no regular use of drugs, health foods, or foods for specified health needs that may affect glucose metabolism; and who did not have excessive smoking or drinking habits. Before the commencement of this study, informed consent was obtained from the subjects in accordance with the Declaration of Helsinki. On the test day, preliminary testing and test food ingestion were conducted in a fasting state (with dinner completed by 8:00 p.m. on the previous day and after fasting for at least 10 h). Furthermore, until completion of all the tests on the test day, the patients were allowed to drink only water, with restrictions on eating and drinking, including gum and candy. Alcohol consumption was prohibited the day before and on the test day. This study was conducted after obtaining approval from the Medical Station Clinic Ethical Review Board (IRB No. 20000022). The study protocol was registered in the Clinical Trials Registration System (UMIN-CTR) (UMIN000041590).

### 2.4. Test Design

The outline of the study design is shown in Figure 1. Participants’ attributes are listed in Table 1. Subjects who provided consent, answered lifestyle questionnaires, participated in the medical interviews, physical measurements, physical examinations, fasting blood and urine collection, and simple blood glucose measurement with a fasting self-monitoring device (Accu-Chek Aviva; Roche DC Japan, Tokyo) were selected as candidate subjects. One of the objectives of this study was to measure the GI of YPP according to ISO 26642:2010; hence, three pretests with 50 g of glucose ingestion were conducted to calculate the area under the ascending curve (IAUC). Subjects with an intra-individual coefficient of variation (CV) of response ≤30% were selected. Fifteen subjects were included in the study based on the preliminary test and were randomly assigned to one of two groups. They were made to consume the test foods in an open crossover design. The intake was conducted in two phases (phases I and II), with a washout period of at least two days in between.

Blood samples were collected immediately before intake and at 15, 30, 45, 60, 90, and 120 min after intake. During the test period, 147 g of rice (corresponding to 50 g of carbohydrate) was consumed 180 min (second meal) from the start of YPP or WP intake (first meal; 0 min) to examine the effect of the second meal on postprandial blood glucose. Blood samples were collected immediately before and at 15, 30, 45, 60, 90, 120, 180, 195, 210, 225, 240, 270, and 300 min after intake. To reduce the pain caused by punctures from repeated blood collection, a cannula was placed in the subject’s vein, and blood was drawn from it. Blood glucose and insulin levels were measured using the hexokinase UV method and chemiluminescence immunoassay (CLIA method) at BML Co.

A questionnaire using a 100 mm Visual Analogue Scale (VAS) was used to evaluate the results at 30, 60, 90, 120, 150, and 180 min from the start of YPP or WP intake. The four questions asked included: (1) how satisfied the subjects felt (about food or eating); (2) how hungry they were; (3) how full they were; and (4) how much they could eat if they were to eat at that moment. The questions were answered on the VAS, where 0 mm and 10 mm indicate the lowest and the highest possible level to quantify satiety and hunger, respectively [18].

The GI of YPP was calculated according to ISO 26642:2010 from blood glucose levels collected immediately before and at 15, 30, 45, 60, 90, and 120 min after consumption, using the following formula:GI value = (IAUC of test food)/IAUC of reference food × 100

### 2.5. Test Food Processing for Postprandial Blood Glucose and Insulin Measurement

Fifty grams of glucose was dissolved in 250 mL of water and used as a reference diet for pretesting. YPP (248 g wet weight) was consumed after it was boiled for 6 min and seasoned with 2 g of dried Shiso seasoning (which contains almost no sugar). For comparison, WP (248 g wet weight) was prepared and consumed the same way as YPP. For the second meal measurement, 147 g of heated rice was weighed and seasoned with 2 g of Shiso seasoning and consumed. In both cases, each bite was chewed about 30 times, and the entire amount was consumed over approximately 10 min.

### 2.6. Component Analysis of the Test Food

The amino acids and nucleic acids of YPP and WP were analyzed using high-performance liquid chromatography (HPLC). Twice the amount of ultrapure water was added to the boiled pasta sample used for the sensory test, and the sample was ground using a homogenizer. It was then centrifuged at 10,000 rpm (9730× *g*), and the supernatant was filtered and subjected to analysis. The glutathione (GSH) content was quantified using Japan Food Research Laboratories. If the product package contained nutritional information, the values were referenced. Dietary fiber contents not listed on the package were analyzed by Japan Food Research Laboratories.

### 2.7. Statistical Analysis

A statistical analysis was conducted using the Excel statistical software Bell Curve for Excel (Social Survey Research Information Co., Ltd., Tokyo, Japan). For the sensory tests, the analysis was performed using an unpaired *t*-test. Corresponding *t*-tests were used to analyze the data for postprandial blood glucose, insulin, and satiety survey measures. A risk rate of less than 5% was considered significant.

## 3. Results

### 3.1. Sensory Evaluation

In the evaluation of pasta, the saltiness score was significantly higher for YPP than for WP at 0.75% salt concentration. The umami score was significantly higher for YPP at 0.75% salinity and only marginally higher than WP at 1.00% salinity. Bitterness and full-bodiedness scores were significantly higher for YPP than for WP at both salinity levels. No significant differences were observed in sweetness, acidity, or overall desirability (Table 1). The evaluation was conducted by 27 staff members who worked for a product manufacturer and who was familiar with the sensory testing of the five basic flavors.

In the evaluation of boiled water, saltiness tended to be higher in YPP than in WP at 0.75% salt concentration. The umami score was significantly higher for YPP than for WP at 0.50% and 0.75% salinity, and only marginally higher than WP at 1.00%. The sweetness score was significantly higher for YPP than for WP at 0.50% and 1.00% salt concentrations. The full-bodiedness score was significantly higher for YPP than for WP at 0.50% and 0.75% salinity and only marginally higher than WP at 1.00%. No significant differences were observed in bitterness, acidity, or overall desirability (Table 2). The evaluation was conducted by 30 staff members who worked for a product manufacturer and was familiar with the sensory testing of the five basic flavors.

### 3.2. Ingredients of Pasta and Boiled Water

To corroborate the results of the sensory testing of YPP and WP, the taste components were analyzed, and the results are shown in Table 3.

The results showed that the total amino acid content in YPP was approximately 14 times higher than that in WP (Table 3). Glutamic acid, an amino acid associated with umami, was >28 times more abundant in YPP. In addition, asparagine, which has a bitter taste, was 28 times more abundant in YPP.

There were no notable differences in sugar content between YPP and WP. Guanylylate, a nucleic acid associated with umami, was significantly higher in YPP than in WP. In terms of organic acids, citric acid was approximately 22 times more abundant in YPP. In addition, GSH could not be detected in WP, but YPP contained a total of 14.01 ± 2.41 mg/100 g.

Boiled water also showed the same tendencies as pasta, with YPP boiled water containing more glutamate, asparagine, guanylate, citric acid, and GSH than WP’s (Table 3).

### 3.3. Postprandial Blood Glucose, Insulin Transition, GI Measurement

Fifteen healthy men and women ate the same weight of YPP or WP as the first meal in the morning and then ate WR as the second meal, and postprandial blood glucose and insulin secretion were measured. The physical characteristics of the participants are presented in Table 4. The nutritional information of the test foods is shown in Table 5. Changes in postprandial blood glucose, insulin secretion, and AUC are shown in Figure 2.

Postprandial blood glucose levels were significantly lower in subjects who ate YPP at 45, 60, 90, 120, and 180 min. In contrast, with YPP consumed as the first meal, the postprandial blood glucose level was significantly higher at 270 min when WR was consumed as a second meal at 180 min (Figure 2A). Insulin levels were significantly lower with YPP treatment at 90, 120, and 180 min. Furthermore, with YPP consumed as the first meal, postprandial insulin levels were significantly lower at 195, 210, 225, and 240 min when WR was consumed as a second meal at 180 min (Figure 2B). The AUC of blood glucose during the first meal was significantly lower when YPP was consumed (Figure 2C). There was no significant difference in the blood glucose AUC of the second meal between subjects who consumed YPP and those who consumed WP for the first meal (Figure 2D). YPP intake was significantly lower than that in both the first and second meals (Figure 2E,F).

The GI of YPP was calculated using the method described in ISO 26642:2010. The GI was 47.0 ± 23.0, indicating that YPP was a low-GI food. However, because WP did not correspond to 50 g of ingested carbohydrates, its GI was not calculated.

### 3.4. Sensations Related to Satiety after Eating

In addition to measuring blood glucose and insulin as described above, 15 healthy men and women also assessed their satisfaction, hunger, and how much more they could eat after the first meal of YPP or WP of the same weight using the VAS method. When comparing YPP and WP of the same weight, there was no significant difference in any of the items (Figure 3).

## 4. Discussion

In this study, we examined the taste characteristics of YPP at different salt concentrations, the maintenance of satiety when consuming the same weight as WP, and the effects on postprandial blood glucose and insulin secretion. This was done to test whether YPP can be a functional staple food contributing to better health.

### 4.1. Taste Properties

To simplify the sensory evaluation of taste in this study, the evaluation was conducted with minimal salt and olive oil seasoning. The results of this study showed that YPP was significantly saltier than WP at the same salt concentration of 0.75%. Although a synergistic effect has been reported in which a salty taste is perceived more intensely in the presence of umami [8], the enhanced salty taste observed in YPP, and its boiled water in this study was considered to be an effect of the umami taste of yellow peas. From the results of the analyses, it was inferred that glutamate and guanylate were responsible for the umami taste of YPP. Interestingly, GSH was detected in YPP, and its concentration was only slightly higher than the previously reported taste threshold (10 mg/100 g) [19]. GSH imparts a full-bodied taste; thus, the result showing that YPP has a significantly more intense full-bodied taste than WP is reasonable.

The search for salty taste substitutes and the development of salty taste enhancers has been actively pursued. Dipeptides with saltiness-enhancing effects have been reported, including dipeptides containing arginine (Arg) [20]. Since we were able to detect GSH in YPP, this effect may be attributable to the saltiness-enhancing peptides contained in yellow peas. It is hoped that further research will identify the causes of these saltiness-enhancing effects.

Salt reduction helps prevent disease. In the UK, a policy aimed at reducing salt content in food was launched in 2003; food companies jointly set voluntary salt reduction targets, especially for processed foods such as bread and cereals [21]. It has been suggested that abruptly reducing the amount of salt or switching to alternatives would ruin the taste of the food and make the salt-reducing efforts short-lived. However, between 2003 and 2011, when salt was gradually reduced so that the change in taste was imperceptible, salt intake in adults was successfully reduced by about 15% from approximately 10 g [22]. As this example shows, efforts to slightly reduce salt intake in staple foods, which make up a large portion of the diet, can be effective. Applying this to this study, for example, if a person consumes 300 g of pasta cooked with 1% salt sauce, he or she would have consumed 3 g of salt per meal, which is more than half of the WHO’s target of 5 g of salt per day. However, because pasta made from YPP with a salt concentration of 0.75% is perceived as slightly saltier, the same 300 g intake can reduce salt intake by 0.75 g per serving. Incorporating staple foods such as YPP into our lives that have characteristics that make them taste slightly saltier than conventional products could help effectively reduce salt intake without compromising taste.

### 4.2. Effects of the First Meal on Blood Sugar, Insulin, and Postprandial Satiety

In this study, the same weights of YPP and WP were consumed as they were in the first meal. Since the amount of carbohydrates ingested was 22 g higher in WP under these conditions, the higher blood glucose and insulin secretion after eating WP as the first meal could be attributed to the higher carbohydrate intake. Rather than reducing food intake to suppress transient increases in blood glucose levels, choosing foods that are more likely to suppress blood glucose elevation and insulin secretion may be more beneficial in maintaining satiety. Maintaining satiety (i.e., feeling less hungry) is important in forming healthy eating habits, because if satiety is not maintained after eating, it can lead to excessive calorie intake through snacking [23]. In this study, the VAS was used to evaluate postprandial satiety-related sensations when the same weight of YPP and WP was consumed as the first meal. There was no difference between the satiety-related sensory values obtained with YPP and WP of the same weight. Furthermore, some studies have shown that low postprandial blood glucose could lead to hunger [24]. However, there was no significant difference in satiety or hunger after eating YPP and WP under the test conditions of this study, despite the significantly less blood glucose elevation observed after YPP consumption. This suggests that when YPP is consumed to control blood glucose levels, hunger is unlikely to lead to overeating, thus, indicating that a staple food processed from yellow peas has an advantage in controlling blood glucose levels.

We compared legumes with other foods and observed that for the same calorie intake, legumes increased satiety by 31% [25]. In this study, yellow pea pasta was compared to wheat pasta of the same weight. Under these conditions, there was no significant difference in satiety, even though wheat pasta had 1.19 times more calories than YPP. This result may be similar to previous studies that have shown legumes to be more satiating than other foods with the same caloric value.

Yellow pea pasta is a staple food rich in dietary fiber and relatively low in carbohydrates. It is a functional staple food because it has better functions, including controlling postprandial blood glucose and insulin and fewer negative factors such as the early onset of hunger. In recent years, brewers’ spent grain extracts, including resistant starch, arabinoxylan, beta-glucan and other soluble fiber, have been shown to lower blood glucose and insulin to normal levels [26]. These components may also contribute to improved glycemic control by YPP. The components and mechanisms involved in improving blood glucose control by YPP are not clear in this study, but if confirmed in long-term studies, they may be used to reduce the development of diabetes and cardiovascular disease.

### 4.3. Effect of Yellow Pea Pasta on the Second Meal

The effect of a previous meal on the degree of blood sugar elevation caused by the next meal has been shown in various foods as a second meal effect. A typical example is barley beta-glucan, which has been shown to reduce the increase in blood sugar after the consumption of the next meal [27]. The YPP used in this study contains 14.7 g of dietary fiber, which is approximately three times the quantity of dietary fiber contained in the same amount of WP (4.5 g). Therefore, in this study, we examined the effects of YPP and WP on blood glucose and insulin levels after the next meal, expecting that this dietary fiber would have a second meal effect. However, we found no significant difference in the blood glucose AUC of the second meal, regardless of what was consumed in the first meal. In contrast, the insulin AUC of the second meal was significantly smaller when YPP was consumed in the first meal. In the YPP group, insulin decreased to the baseline level at 180 min after the meal. In contrast, insulin did not decrease to the baseline level in the WP group even at 180 min, suggesting that the insulin AUC was much greater. Focusing not only on blood glucose but also on insulin, and choosing YPP as a staple food might contribute to the maintenance of insulin at low levels.

This study had some limitations. First, the sensory tests of YPP-like foods were conducted only by Japanese panelists; thus, the possibility that there were biases associated with Japanese eating habits and preferences cannot be denied. Second, since hunger, blood glucose, and insulin measurements were also taken only in Japanese individuals, we cannot deny the possibility that racial differences may have caused a bias in the responses. Further research on the preferences of YPP-like foods and the blood glucose and insulin responses to YPP-like foods in regions and races other than Asia is desirable. Hence, they can be incorporated into diets as bean-based staples in various parts of the world. Third, the phenomenon that YPP was perceived to be saltier than WP at 0.75% salt concentration was only the result of a single experiment with only one type of seasoning. Thus, we cannot generalize whether these effects are likewise observed with YPP seasoned with tomatoes or cream, which are the more typical ingredients used in pasta. The next challenge would be to verify the saltiness-enhancing and blood sugar-controlling benefits of consuming YPP prepared using the conventional cooking methods for pasta.

## 5. Conclusions

In this study, pasta made from yellow pea tasted saltier than conventional pasta made from semolina wheat at 0.75% salt concentration, suggesting that it may be suitable for eating at lower salt concentrations. Furthermore, the GI was confirmed to be 47. There was no difference in how well the same weight of conventional pasta and pasta made from yellow pea maintained satiety. However, blood sugar and insulin secretion were significantly lower, and insulin secretion at the next meal was lower when YPP was consumed. Thus, this confirmed that YPP is a staple food that is advantageous for blood sugar control. Based on the above, we concluded that pasta made from yellow peas is a functional staple food that may help reduce salt intake and control blood sugar.

## Figures and Tables

**Figure 1 nutrients-15-00283-f001:**
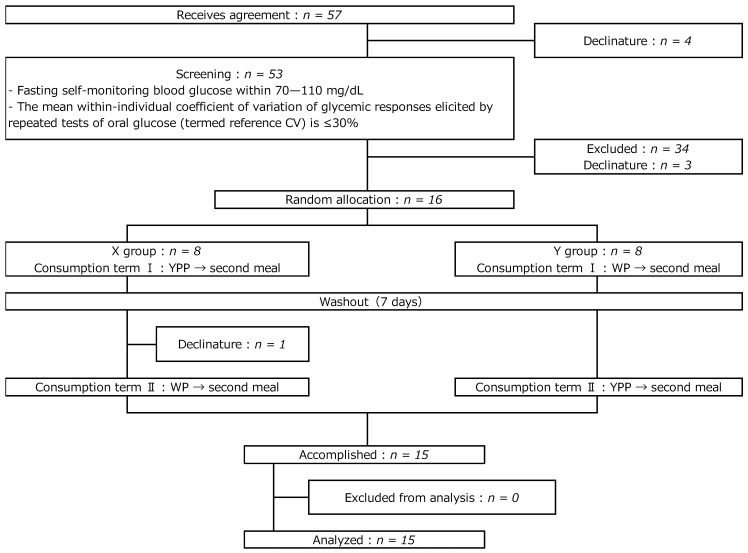
Study flow diagram.

**Figure 2 nutrients-15-00283-f002:**
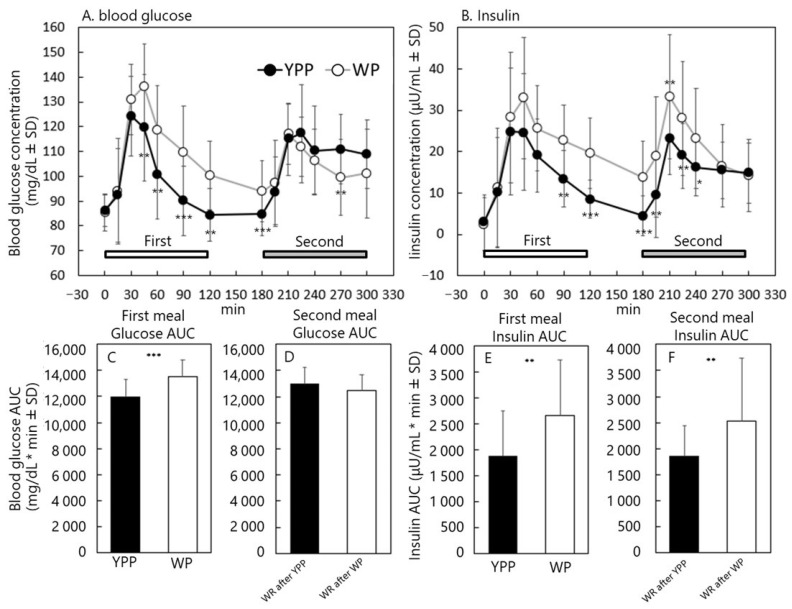
Changes in postprandial blood glucose, insulin secretion, and AUC. Blood glucose (**A**) and insulin (**B**) readings at each elapsed time of the first and second meals. AUC of blood glucose (**C**) and AUC of insulin (**D**) at first meal. AUC of blood glucose (**E**) and AUC of insulin (**F**) at second meal * *p* value < 0.05. ** *p* value < 0.01. *** *p* value < 0.001.

**Figure 3 nutrients-15-00283-f003:**
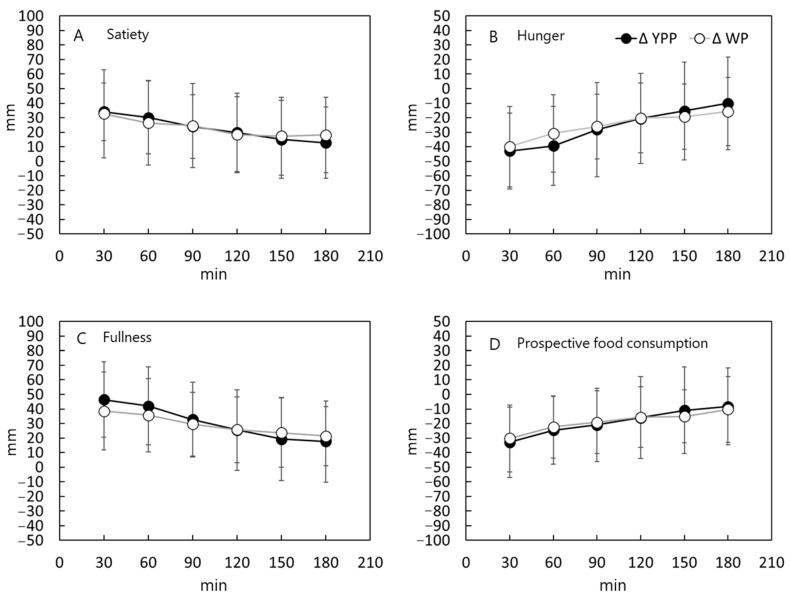
Assessment of satisfaction, hunger, and how much more individuals could eat after the first meal of YPP or WP of the same weight using the VAS method. A questionnaire using a 100 mm Visual Analogue Scale (VAS) was used to evaluate the results at 30, 60, 90, 120, 150, and 180 min from the start of YPP or WP intake. The four questions asked included: (**A**) how satisfied the subjects felt (about food or eating); (**B**) how hungry they were; (**C**) how full they were; and (**D**) how much they could eat if they were to eat at that moment.

**Table 1 nutrients-15-00283-t001:** Sensory evaluation results of pasta samples.

	Salinity	YPP	WP	*p* Value
saltiness	0.50%	4.56 ± 1.05	4.59 ± 1.60	0.920
	0.75%	5.57 ± 1.35	4.54 ± 1.55	0.010 *
	1.00%	6.18 ± 1.49	5.89 ± 1.31	0.450
umami	0.50%	5.56 ± 1.28	5.15 ± 1.38	0.266
	0.75%	6.43 ± 1.07	5.18 ± 1.16	0.000 ***
	1.00%	6.04 ± 1.37	5.39 ± 1.13	0.062 †
sweetness	0.50%	5.15 ± 1.29	5.07 ± 1.69	0.857
	0.75%	5.21 ± 1.40	4.82 ± 1.70	0.349
	1.00%	5.29 ± 1.41	4.79 ± 1.57	0.216
bitterness	0.50%	4.78 ± 2.04	2.74 ± 1.58	0.000 ***
	0.75%	3.82 ± 2.11	2.39 ± 1.57	0.005 **
	1.00%	4.25 ± 2.19	2.86 ± 1.64	0.008 **
sourness	0.50%	3.89 ± 1.80	3.22 ± 1.78	0.178
	0.75%	2.93 ± 1.92	3.11 ± 1.61	0.721
	1.00%	3.39 ± 1.87	3.39 ± 1.39	1.000
kokumi	0.50%	5.59 ± 1.42	4.63 ± 1.45	0.017 *
	0.75%	5.82 ± 1.12	4.64 ± 1.57	0.002 **
	1.00%	5.57 ± 1.40	4.54 ± 1.64	0.014 *
overall acceptance	0.50%	5.11 ± 1.74	5.70 ± 1.14	0.145
	0.75%	5.96 ± 1.62	5.68 ± 1.61	0.511
	1.00%	5.68 ± 1.44	5.64 ± 1.39	0.925

All the value are mean ± SD of 27 individual determinations. * *p* value < 0.05. ** *p* value < 0.01. *** *p* value < 0.001. † *p* value < 0.1.

**Table 2 nutrients-15-00283-t002:** Sensory evaluation results of hot water extracts samples.

	Salinity	YPP-ex	WP-ex	*p* Value
Saltiness	0.50%	4.67 ± 0.92	4.47 ± 1.31	0.496
	0.75%	5.53 ± 0.82	5.13 ± 0.97	0.091 †
	1.00%	5.73 ± 0.98	5.67 ± 1.03	0.798
Umami	0.50%	5.13 ± 1.14	4.50 ± 0.97	0.024 *
	0.75%	5.30 ± 1.18	4.63 ± 0.89	0.017 *
	1.00%	5.20 ± 1.30	4.63 ± 1.10	0.073 †
Sweetness	0.50%	4.87 ± 1.17	4.17 ± 0.99	0.015 *
	0.75%	4.60 ± 1.28	4.23 ± 1.33	0.281
	1.00%	4.87 ± 1.07	4.27 ± 1.05	0.033 *
Bitterness	0.50%	4.27 ± 1.44	4.03 ± 1.33	0.516
	0.75%	4.60 ± 1.52	4.07 ± 1.44	0.168
	1.00%	4.23 ± 1.43	4.03 ± 1.54	0.605
Sourness	0.50%	3.97 ± 1.40	3.93 ± 1.26	0.923
	0.75%	4.27 ± 1.23	4.23 ± 1.17	0.915
	1.00%	4.13 ± 1.33	3.87 ± 1.43	0.458
Kokumi	0.50%	5.27 ± 1.26	4.23 ± 1.25	0.002 **
	0.75%	5.43 ± 1.45	4.80 ± 1.30	0.080 †
	1.00%	5.63 ± 1.19	4.87 ± 1.59	0.039 *
overall acceptance	0.50%	4.67 ± 1.37	4.23 ± 1.14	0.188
	0.75%	5.23 ± 1.30	4.97 ± 1.38	0.444
	1.00%	5.27 ± 1.57	4.80 ± 1.54	0.251

All the value are mean ± SD of 30 individual determinations. * *p* value < 0.05. ** *p* value < 0.01. † *p* value < 0.1.

**Table 3 nutrients-15-00283-t003:** Comparison of non-seasoned YPP, YPP-ex, CP, and CP-ex components.

Amino Acids (mg/100 g)	YPP	YPP-ex	WP	WP-ex
Aspartic Acid	8.70 ± 0.19	2.23 ± 0.01	1.73 ± 0.00	0.70 ± 0.00
Threonine	1.53 ± 0.03	0.34 ± 0.00	0.22 ± 0.01	0.13 ± 0.01
Serine	2.69 ± 0.01	0.60 ± 0.00	0.25 ± 0.00	0.11 ± 0.00
Glutamic Acid	42.78 ± 0.79	10.20 ± 0.02	1.64 ± 0.01	0.76 ± 0.00
Proline	2.14 ± 0.05	0.42 ± 0.01	0.50 ± 0.02	0.24 ± 0.00
Glycine	2.48 ± 0.04	0.66 ± 0.00	0.34 ± 0.00	0.25 ± 0.03
Alanine	2.11 ± 0.03	0.50 ± 0.00	1.19 ± 0.00	0.50 ± 0.00
Valine	1.58 ± 0.02	0.30 ± 0.00	0.28 ± 0.00	0.15 ± 0.00
Cysteine	N.D. (not detected)	N.D.	0.11 ± 0.00	0.07 ± 0.00
Methionine	0.95 ± 0.02	0.15 ± 0.00	0.04 ± 0.00	0.03 ± 0.00
Isoleucine	0.62 ± 0.03	0.16 ± 0.00	0.12 ± 0.00	0.07 ± 0.00
Leucine	0.94 ± 0.01	0.02 ± 0.01	0.21 ± 0.00	0.13 ± 0.00
Tyrosine	1.50 ± 0.10	0.24 ± 0.01	0.26 ± 0.01	0.18 ± 0.00
Phenylalanine	2.22 ± 0.03	0.35 ± 0.01	0.10 ± 0.00	0.07 ± 0.00
Histidine	1.09 ± 0.02	0.19 ± 0.00	0.08 ± 0.00	0.04 ± 0.00
Lysine	4.79 ± 0.06	0.45 ± 0.00	0.18 ± 0.00	0.13 ± 0.00
Tryptophan	1.91 ± 0.06	0.31 ± 0.01	1.44 ± 0.01	0.99 ± 0.02
Arginine	65.47 ± 0.01	6.13 ± 0.03	0.47 ± 0.01	0.34 ± 0.00
Asparagine	25.96 ± 0.50	5.66 ± 0.01	2.64 ± 0.02	1.18 ± 0.00
Glutamine	0.64 ± 0.04	0.14 ± 0.01	0.10 ± 0.00	0.03 ± 0.00
Total amino acids	170.11 ± 3.42	29.13 ± 0.14	11.89 ± 0.11	5.99 ± 0.09
Sugars (g/100 g)				
Fructose	N.D.	N.D.	0.02 ± 0.00	0.01 ± 0.00
Glucose	N.D.	N.D.	0.02 ± 0.01	0.01 ± 0.00
Sucrose	0.39 ± 0.01	0.10 ± 0.00	0.05 ± 0.01	0.04 ± 0.01
Maltose	N.D.	N.D.	0.17 ± 0.00	0.15 ± 0.01
Total sugars	0.39 ± 0.01	0.10 ± 0.00	0.26 ± 0.02	0.21 ± 0.02
Flavour nucleotides (mg/100 g)				
inosinic acid	N.D.	N.D.	N.D.	N.D.
Guanilic acid	9.98 ± 0.23	3.07 ± 0.03	0.01 ± 0.00	0.03 ± 0.02
Organic acids (mg/100 g)				
Citric acid	98.00 ± 0.84	38.80 ± 0.09	4.30 ± 0.05	2.50 ± 0.27
Malic acid	11.52 ± 0.21	6.84 ± 0.02	14.13 ± 0.03	5.41 ± 0.50
Succinic acid	0.56 ± 0.00	0.32 ± 0.00	1.59 ± 0.01	0.57 ± 0.05
Lactic acid	1.01 ± 0.06	0.70 ± 0.05	0.39 ± 0.06	0.48 ± 0.10
Acetic acid	1.89 ± 0.14	1.09 ± 0.02	1.03 ± 0.01	0.75 ± 0.06
Total organic acids	112.98 ± 1.25	47.75 ± 0.18	21.44 ± 0.15	10.07 ± 0.98
Gurutation (mg/100 g)				
Glutathione (Reduced Form)	9.01 ± 1.77	2.44 ± 0.36	N.D.	N.D.
Glutathione (Oxidized Form)	5.00 ± 0.63	3.35 ± 0.47	N.D.	N.D.
Total glutathione	14.01 ± 2.41	5.79 ± 0.83	N.D.	N.D.

**Table 4 nutrients-15-00283-t004:** Descriptive characteristics of the study subjects.

Characteristic	Mean ± SD
Age (yrs)	40.2 ± 6.8
Weight (kg)	55.7 ± 8.3
Height (cm)	163.5 ± 8.4
BMI (kg/m^2^)	20.8 ± 1.7
Systolic blood pressure (mmHg)	110.6 ± 10.3
Diastolic blood pressure (mmHg)	66.7 ± 6.8
Fasting blood glucose (mg/dL)	85.8 ± 4.6
Triglyceride (mg/dL)	70.3 ± 23.3
Total cholesterol (mg/dL)	198.5 ± 31.5
HDL-C (mg/dL)	68.5 ± 12.8
LDL-C (mg/dL)	112.4 ± 27.0

**Table 5 nutrients-15-00283-t005:** Nutritional composition of test foods.

Nutritional Composition	Glucose (Reference)	YPP	WP	White Rice
Energy (kJ) ^a^	850	1371	1637	917
Protein (g)	0	19.7 †	12.9 †	3 †
Fat (g)	0	1.9 †	2 †	0 †
Total dietary fiber (g)	0	14.7 †	4.5	2 †
Available carbohydrate (g)	50 †	50 †	77 †	50 †
Consumption weight (g)	300 *	248	248	147

† Referenced from product nutrition facts. ^a^ Calculated as protein = 17 kJ/g, fat = 37 kJ/g, available carbohydrate = 17 kJ/g, and total dietary fiber = 8 kJ/g. * Dissolved in 250 mL of water.

## Data Availability

The data that support the findings of this study are available from the corresponding author, upon reasonable request.

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
