# Peer review of "Yellow Pea Pasta Enhances the Saltiness and Suppression of Postprandial Blood Glucose Elevation"

_nutrients, 2023, doi:10.3390/nu15020283_

Round 1

Reviewer 1 Report

In this study, Tsuchiya et al. investigate the perceived saltiness of yellow pea pasta and its effect on postprandial glucose and insulin levels and postprandial hunger in comparison to equal weight wheat pasta.

I only have some minor comments : 

Title

Please try to give a more precise title. Also revise "postpriandal" to "postprandial".

Abstract 

Line 16 : Maybe consider replacing "inhibited" with "lower".  

Introduction

Line 77 : please revise

Line 80 : please revise

Results

Figure 2 : Please explain A, B, C, D, E and F in Figure's 2 legend. Give titles to axes. 

Figure 3: Please revise "assesstion" and give titles to axes.

References

Please revise the Reference list according to Journal's instructions for authors.

Reviewer 2 Report

Comments:

Section 1: Abstract:

Q1. This article is interesting, but the expressions need to be further improved. “we further examined its effectiveness” is not a persuasive reason for further research. The research significance and necessity are suggested to be supplemented.

Section 2: Introduction:

Q2. The excessive sodium chloride (salt) intake is wellknown to increase the risk of hypertension, cardiovascular disease, and kidney disease, as the authors stated, but did any papers report the negative effects on blood sugar control? If possible, the relevant expressions need to be added (mainly demontrated) and the references need to be cited.

Q3: Introduction of Yellow Pea Pasta” in recent three years was suggested to be supplemented, including the chemical constituents and mechanisms associated with hypoglycemia.

Q4: The last paragraph of Introduction section needed to be further improved for better presenting the research significance and innovation.

Section 3: Materials and Methods:

Q5. In table 1, although the overall acceptance of YPP and WP showed no significant differences, but would the other sensory evaluation indicators with significant difference indicate exitence of some active ingredients, thereby affecting the blood glucose control?

Section 4: Results:

Q6. Table 2, “the saltiness score was significantly higher for YPP than for WP at 0.75% salt concentration”, but I found the p value was 0.091? as well as the subsequent 0.073, 0.080? Dose the “A risk rate of less than 5% was considered significant” mean “p <0.05 was considered significant”.

Q7. All symbols in the table need to be explained in the following notes, such as “*”, ”**”, ””.

Q8. Table 3, did the authors determined the components as free components? As is known to us, proteins or polysaccharides could not well present the characteristics of amino acids,monosaccharides, etc.

Q9. Did the authors consider active ingredients contents and their functions with high molecular weignts including proteins or polysaccharides?

Section 5: Discussion:

Q10. To my knowledge, there are two main action mechanisms of hypoglycemic food: 1. There are substances in food that can inhibit the activity of insulin or the activity of glucosidase, which can take effect after oral administration and reduce the rate of blood glucose rise; 2, the starch content in food decreases, and the content of non-starch substances (protein, cellulose, etc.) increases, making the body unable to produce a large amount of glucose into the blood. It is suggested that the authors discuss refer to these two aspects (or more persuasive perspectives) to improve the depth of this study.

Section 6: References:

Q11. The relevant papers especially in the past three years was lacked and the reference formats did not meet the requirements of nutrients.

Reviewer 3 Report

Thanks to submit the manuscript "Functions of Yellow Pea Pasta: Intenser Perceived Saltiness and

Protection from Postpriandal Blood Glucose Elevation" to Nutrients. 

Overall the subject of the manuscript is interesting and falls within the scope of Nutrients. However, the text still needs to be organized for clarity.

Abstract: needs to be reformulated because it is not possible to understand what was done in the work. it is very confusing, for example the work was done with pea and in the last lines there is information about rice.

Line#75: space

Line#77: It is more common for the objective to be just a sentence. please rephrase.

Line#91: only one. expert was used? it is not common, even if it is in cases of people with a lot of experience, that only one person is used in the sensory evaluation. to validate this sensorial it would be interesting that more people were used.

Lime#154: what is VÁS?

Line#164: Is this table in the right place?

Line#191: noodles or pasta?

Line#200: "higher" Needs "than"

Line#203: if only one expert was used, how come there is standard deviation in this table? explain this in the topic Statistical Analysis.

Figure 2: The legend must contain what each graphic means as well as its abbreviations. impossible to understand as it is.

Line#250: arrange this text after the figure caption

Round 2

Reviewer 2 Report

Thanks for these careful modifications, I have no any other questions.